# Histologic Analysis of Idiopathic Pulmonary Fibrosis by Morphometric and Fractal Analysis

**DOI:** 10.3390/biomedicines11051483

**Published:** 2023-05-19

**Authors:** Massimiliano Mancini, Lavinia Bargiacchi, Claudia De Vitis, Michela D’Ascanio, Chiara De Dominicis, Mohsen Ibrahim, Erino Angelo Rendina, Alberto Ricci, Arianna Di Napoli, Rita Mancini, Andrea Vecchione

**Affiliations:** 1Morphologic and Molecular Pathology Unit, Sant’Andrea University Hospital, 00189 Rome, Italy; 2Department of Clinical and Molecular Medicine Sant’Andrea University Hospital, “Sapienza” University of Rome”, 00189 Rome, Italyalberto.ricci@uniroma1.it (A.R.);; 3UOC Respiratory Disease, Sant’Andrea University Hospital, 00189 Rome, Italy; 4Radiology Unit, Sant’Andrea University Hospital, 00189 Rome, Italy; 5Thoracic Surgery Unit, Sant’Andrea University Hospital, “Sapienza” University of Rome, 00189 Rome, Italy

**Keywords:** IPF, UIP, fractals, histomorphometry

## Abstract

Idiopathic pulmonary fibrosis (IPF) is a chronic, progressive fibrotic lung disorder, ultimately leading to respiratory failure and death. Despite great research advances in understanding the mechanisms underlying the disease, its diagnosis, and its treatment, IPF still remains idiopathic without known biological or histological markers able to predict disease progression or response to treatment. The histologic hallmark of IPF is usual interstitial pneumonia (UIP), with its intricate architectural distortion and temporal inhomogeneity. We hypothesize that normal lung alveolar architecture can be compared to fractals, such as the Pythagoras tree with its fractal dimension (*D_f_*), and every pathological insult, distorting the normal lung structure, could result in *D_f_* variations. In this study, we aimed to assess the UIP histologic fractal dimension in relationship to other morphometric parameters in newly diagnosed IPF patients and its possible role in the prognostic stratification of the disease. Clinical data and lung tissue specimens were obtained from twelve patients with IPF, twelve patients with non-specific interstitial pneumonia (NSIP), and age-matched “healthy” control lung tissue from patients undergoing lung surgery for other causes. Histology and histomorphometry were performed to evaluate *D_f_* and lacunarity measures, using the box counting method on the FracLac ImageJ plugin. The results showed that *D_f_* was significantly higher in IPF patients compared to controls and fibrotic NSIP patients, indicating greater architectural distortion in IPF. Additionally, high *D_f_* values were associated with higher fibroblastic foci density and worse prognostic outcomes in IPF, suggesting that *D_f_* may serve as a potential novel prognostic marker for IPF. The scalability of *D_f_* measurements was demonstrated through repeated measurements on smaller portions from the same surgical biopsies, which were selected to mimic a cryobiopsy. Our study provides further evidence to support the use of fractal morphometry as a tool for quantifying and determining lung tissue remodeling in IPF, and we demonstrated a significant correlation between histological and radiological *D_f_* in UIP pattern, as well as a significant association between *D_f_* and FF density. Furthermore, our study demonstrates the scalability and self-similarity of *D_f_* measurements across different biopsy types, including surgical and smaller specimens.

## 1. Introduction

Idiopathic pulmonary fibrosis (IPF) is a chronic and progressive fibrosing interstitial lung disease (ILD), with a poor prognosis and survival rate. IPF diagnosis relies on a multidisciplinary approach, combining clinical, radiological, and pathological data [1]. During the last few years, the widespread use of CT (Computed Tomography) radiology has led to an improved detection of the disease, with improved awareness and with greater understanding of the pathogenesis leading to the development of multiple therapeutic targets. IPF patients are at greater risk of developing lung cancer [2]. IPF assumes many similarities with cancer and is characterized by both genetic and epigenetic dysregulation, with abnormal activation of transduction pathways and uncontrolled cell cycle proliferation [3,4,5,6]. The IPF prognosis is grim, with half of those diagnosed, and they do not receive any antifibrotic treatment, which is able to slow down the progression of the disease, succumbing to respiratory failure within three to five years. Pulmonary function tests are now the main predictor of the survival of progression of the disease [7].

Currently, computer-based assessment of fibrosis on CT offers an accurate and consistent way to measure disease progression and predict mortality, effectively determining the extent and advancement of the condition [8,9]. Even though UIP has proven to be a clinically significant pathologic diagnosis [10,11] thus far, only a limited number of histological factors have been established as predictors; among these, fibroblast foci (FF) play a crucial role in IPF, with many studies showing a high number of FFs, correlating with poor outcomes; conflicting findings have also been reported [12,13,14]. Although basic histologic (and radiologic) measurements of fibrosis areas and volumes are widely used, these techniques partially fail to account for the intricate architectural distortion and temporal inhomogeneity of IPF lungs.

Mandelbrot stated that natural objects, such as trees, mountains, coastlines, and clouds, do not conform to simple geometric shapes, such as cones, pyramids, straight lines, or spheres. Similarly, there is a lack of objectivity in historical accounts when describing the organs and structures of the human body. Fractal objects possess a crucial characteristic, whereby the patterns defining them are consistently replicated at decreasing magnitudes. This leads to their constituent parts, exhibiting a likeness to the whole structure, irrespective of their dimensions; this feature is commonly known as “self-similarity”. We hypothesize that normal lung alveolar architecture can be compared to fractals, such as the Pythagoras tree, with its fractal dimension (*D_f_*), and every pathological insult distorting the normal lung structure could result in *D_f_* variation. Moreover, fractal objects possess a crucial characteristic, whereby the patterns defining them are consistently replicated at decreasing magnitudes, and this leads to their constituent parts, which exhibit a likeness to the whole structure, irrespective of their dimensions: this feature is commonly known as “self-similarity”. Many anatomical structures have been shown to demonstrate self-similarity properties [15], and lungs, with their bronchioalveolar ramifications, can be one of them. Such an approach has been successfully used by radiologists to characterize the spatial patterns of disease progression in ILD, and we believe that this approach, implemented using digital pathology, could also be successful at the histological level, with fractal morphometry assessing architectural distortion and correlating with a specific pattern, thus enforcing histologic pattern recognition [16].

Our study aimed to assess histologic fractal dimensions in connection to other morphometric parameters in newly diagnosed IPF patients and their relation to prognostic stratification of the disease.

## 2. Materials and Methods

### 2.1. Clinical Data

From January 2018 to December 2022, we studied 12 patients who underwent surgery for lung neoplasia without a prior history or symptoms of ILD. The UIP pattern on CT scans was an accidental finding in the diagnostic work-up for lung neoplasia; clinical data, including spirometric examination with DLCO (diffusing capacity of the lungs for carbon monoxide) evaluation, were acquired. All cases were reviewed by multidisciplinary discussion according to the latest guidelines [1,17], and the patients were sent for surgery after careful global evaluation. The lobectomy or wedge resection specimens were formalin-fixed by airway inflation and examined by a dedicated pulmonary pathologist. Paired age-matched control lung tissue was obtained from patients (n = 12) undergoing lung surgery for other causes (i.e., wedge resection for emphysema). Age-matched (n = 12) patients with fibrotic NSIP were also retrieved from our archive to compare histological lung morphometry parameters.

### 2.2. Histology and Histomorphometry

Four to five blocks of peripheric/subpleural pulmonary tissue unaffected by neoplasia with underlying interstitial damage were obtained from each specimen. The lung tissue was paraffin-embedded, and multiple 2 mm thick sections were cut, and these sections were hematoxylin/eosin-stained for microscopic examination. Additional sections were immune-stained for Ki67 (MiB1 clone, Dako-Agilent Technologies, Santa Clara, CA, USA) to evaluate the proliferative index of fibroblastic foci (FF). Representative sections from each specimen (1–2) were acquired with a digital slide scanner (Aperio CS2, Leica Biosystems Nussloch GmbH, Nußloch, Germany) and stored for morphometric evaluation, and UIP diagnosis was established according to ATS/ERS 2018–2022 guidelines after correlation between histologic and radiologic data [1,17] (Appendix A). Morphometrical analyses of whole section images, from both cases and controls, were performed using ImageJ software v. 1.52 (National Institute of Health, Bethesda, MD, USA) to measure the following parameters: (i) FF total number per section; (ii) FF density; and (iii) FF ki-67 index. FF were identified as regions located beneath the pneumocyte layer, where fibroblasts and myofibroblasts were aligned in a straight pattern within a lightly colored newly formed extracellular matrix. *D_f_* and lacunarity measures were obtained after channel separation and green channel thresholding, using the box counting method on the FracLac ImageJ plugin (Karperien, A., FracLac for ImageJ (v.1.54B) http://rsb.info.nih.gov/ij/plugins/fraclac/FLHelp/Introduction.htm. 1999–2013 accessed on 1 August 2022). *D_f_* assessment was performed on both scanned slides and the corresponding regions of interest from the CT slices. Control specimens from healthy individuals, undergoing lung surgery for benign conditions, were acquired and measured. To better understand the value of *D_f_* and its relation to parenchymal architectural distortion on ILD, we also analyzed the same parameters on 12 cases of fibrous non-specific interstitial pneumonia specimens (NSIP) from our archive. To assess *D_f_* self-similarity, prognostic value, and reproducibility of non-surgical specimens, we repeated the measurements on smaller portions from the same surgical biopsies, reducing the region of interest (ROI) to a size, grossly corresponding to a transbronchial cryobiopsy (~36 mm^2^). To further improve our analyses and to make it closer to a real clinical scenario, we used a random number generator set from 10 to 36 to mimic criobiopsy area surface, and we subsequently segmented the histologic section into multiple ROIs and ran the FracLac plugin on each of these subsections. Data were recorded, and the mean *D_f_* was compared to those previously measured on the whole section. Morphometrical analyses were carried out by a dedicated pathologist who performed all the measurements blinded to group assignment.

### 2.3. Statistics

Data were analyzed in GraphPad Prism 9. All results are expressed as the mean ± standard deviation. Analysis of variance was obtained by a two-way nonparametric ANOVA test. The measure of linear correlation between the sets of data was expressed as the Pearson correlation coefficient.

## 3. Results

In this study, we enrolled 12 patients (M:11; F:1; mean age 73 ± 5.071, max 80, min 64) with comprehensive clinical data recorded prior to surgery (Table 1).

After a 36 months follow-up, nine of the twelve patients died because of respiratory failure unrelated to pulmonary neoplasia (two of them because of acute exacerbations). Histological examination was performed by a dedicated lung pathologist, and biopsies were categorized according to ATS/ERS criteria [17]. A final diagnosis of IPF was established on multidisciplinary grounds for each of the patient (For a comprehensive summary, please refer to the correlation matrix in Figure 1a).

### 3.1. UIP and D_f_ Analysis

Patients were divided into histopathology categories of suspects, according to patterns and features [17]. Five out of 12 patients were histologically diagnosed with UIP definite patterns (Table 2), four were diagnosed with probable UIP patterns, and the remaining three were diagnosed as indeterminate for UIP patterns. A final diagnosis of IPF, correlating histological and radiological data, was achieved. We observed no significant correlation between UIP patterns and lung cancer histology of each patient.

We also observed no significant correlation between the UIP pattern histologic sub-categories and spirometric data. A UIP definite pattern showed a reduced survival time compared to probable UIP and indeterminate UIP patients (8.2 vs. 25 and 26.33 months respectively, *p* < 0.05) (Figure 2a). The mean histologic *D_f_* in UIP patients was 1.75 (min. 1.70, max 1.79; std. dev 0.03), and the *D_f_* in the UIP certain pattern was significantly different compared to probable and alternative patterns (1.78 ± 0.01 vs. 1.72 ± 0.02, *p* < 0.0001) (Figure 2b). Patients with a IPF-UIP pattern showed a statistically higher significant difference between *D_f_* compared to normal controls (1.53 ± 0.06, *p* < 0.0001) and patients with NSIP (1.57 ± 0.05; *p* < 0.0001). Interestingly, no significant *D_f_* difference was observed between the NSIP and the control group (see Figure 2c), thus confirming the architectural distortion underlying the pathogenesis of UIP; fibrotic tissue quantification by area surface occupancy did not show significant differences between UIP and NSIP patients.

*D_f_* was inversely correlated with DLCO (−0.77, *p* < 0.005) and positively correlated with follow-up time (+0.63 *p* < 0.05). Histologic *D_f_* positively correlated with radiologic *D_f_* (+0.58, r^2^ = 0.34 *p* < 0.05 Figure 3a). Radiologic *D_f_* in the histologic UIP certain pattern was significantly different compared to the histologic UIP indeterminate sub-category (1.36 vs. 1.19 *p* > 0.05 Figure 3b).

Histologic *D_f_* inversely correlated with follow-up (−0.87, *p* < 0.001). Lacunarity measurements did not correlate with other histological or clinical data.

Survival analysis showed a *D_f_* > 1.75 to detect a category of patients with worse prognosis and overall survival <18 months; the same category of patients showed FF consistently higher compared to those with a lower *D_f_* (Figure 4).

Among the two patients who died of acute exacerbations, both had high *D_f_* measurements (1.796 and 1.794), suggesting a possible link between the degree of complexity of lung architectural distortion and the risk for acute exacerbations.

### 3.2. FF Density and D_f_ Analysis

FF density showed good correlation with histologic *D_f_* (+0.74 *p* < 0.005) and an inverse correlation with follow-up time (−0.73, *p* < 0.005); FF density was significantly higher in certain UIP patterns compared to other patterns (4.004 ± 2.68 vs. 1.49 ± 0.83, *p* < 0.05). We observed no significant correlation between FF density and spirometric data, and no significant difference was observed in the KI67 proliferation index and total FF number.

### 3.3. D_f_ Measurements Scalability and Self-Similarity

*D_f_* measurement comparison between surgical biopsy specimens and ROI, corresponding to hypothetical cryobiopsy surfaces, showed an almost perfect correlation (+0.97, r^2^ = 0.94, *p* < 0.0001 Figure 5a,b) with retention of all correlation and survival parameters.

## 4. Discussion

In this study, we investigated the fractal morphometry of lung tissue in patients affected by idiopathic pulmonary fibrosis (IPF) with a focus on the usual interstitial pneumonia (UIP) pattern. Our results demonstrate a significant correlation between histological and radiological fractal dimensions (*D_f_*) in patients with the UIP pattern, which suggests a consistent architectural distortion underlying the pathogenesis of the disease. We also observed that *D_f_* measurements are scalable and self-similar across different biopsy types, including surgical and cryo-biopsy specimens. Additionally, we found a significant association between *D_f_* and fibroblast foci (FF) density, which is a known predictor of disease progression in IPF.

To our knowledge, this is the first work dealing with fractal morphometry and the usual interstitial pneumonia pattern in the lung. UIP diagnosis can be difficult, especially at present, given the increasing accuracy in CT diagnosis and only problematic cases undergoing biopsy. Nevertheless, histologic diagnosis remains the gold standard for UIP diagnosis [17]. The widely accepted [16] pattern recognition-based approach here meets a numeric quantification strictly associated with architectural distortion and texture analysis of lung parenchyma, providing a quantitative measurement of the complexity underlying the disease. The fractal dimension has been used in other medical branches to quantify difficult-to-describe biological parameters, such as cortical complexity [18], to estimate the complexity of the vascular network [19,20]. Moreover, a role has emerged in detecting changes in tumor vascularity, as well as chromatin shaping, as well as other cancer-related structural alterations at the histological level [21].

In our study, we have demonstrated different levels of complexity characterizing ILD, and, in particular, we have shown a higher degree of distortion in UIP patients compared to fibrotic NSIP patients with similar amounts of total fibrosis. Given the limitations of quantifying such a difficult parameter, the ability to recognize UIP definite histologic patterns, relating IPF patients to poor prognosis, highlights the importance of the histopathologic diagnosis [10,22,23].

In patients with a definite UIP pattern, *D_f_* is higher compared to other patterns, and observed survival curves are observed for patients with definite UIP, who have lower survival rates. The presence of increased *D_f_* can be considered a prognostic factor for the disease. Additionally, an increase in *D_f_* is mainly found in patients with IPF, rather than normal or NSIP patients, indicating a strong specificity in reflecting the disease.

In addition to *D_f_*, our study also investigated the relationship between FF density and disease progression in IPF. FFs are aggregates of activated fibroblasts that are thought to play a key role in the pathogenesis of IPF by driving aberrant tissue remodeling [24,25,26]. Our finding of a significant correlation between FF density and *D_f_* is consistent with previous studies that have shown that FFs are a strong predictor of disease progression in IPF [14,27,28]. Our study adds to this body of the literature by demonstrating that FF density is also correlated with disease progression, as measured by follow-up time. Interestingly, the study did not find any significant correlation between FF density and spirometric data or the KI67 proliferation index and total FF number. This suggests that FF density may be a more specific indicator of certain pathological processes in the lungs, rather than a general marker of lung dysfunction and cell proliferation.

Importantly, we also demonstrated the scalability and self-similarity of *D_f_* measurements across different biopsy types, including surgical and cryobiopsy specimens. This finding has important implications for clinical practice, as it suggests that *D_f_* measurements can be reliably compared across different biopsy types and imaging modalities. This is particularly relevant for cryobiopsy, which has emerged as a promising alternative to surgical biopsy for the diagnosis of IPF [29]. Our study provides further evidence to support the use of cryobiopsy for the diagnosis and monitoring of IPF, even though contrasting reports on the diagnostic yield have been discussed [30,31]. We believe this additional finding is very important because it suggests that cryobiopsies, which are obtained from a less invasive procedure than surgical biopsy, may be a suitable alternative to gather tissue samples for diagnostic or research purposes, without compromising the diagnostic accuracy, the feasibility of the measurements, or the ability to predict patient outcomes. We prefer to stress the advantage of *D_f_* in smaller specimens, showing a higher yield compared to FF, which can be unevenly distributed in the lung parenchyma and unsampled in bioptic specimens, thus reducing their prognostic utility.

Given the rising availability and lower pricing of digital slide scanners, the use of fractal histomorphometry could be easily added in the diagnostic work-up of both surgical and small biopsy samples. It could be particularly useful in differential diagnosis of difficult cases and could help the clinician in the follow-up of worse prognosis patients. Moreover, it could be implemented in artificial intelligence-guided pattern recognition, helping to provide a more comprehensive approach for difficult cases.

Despite these promising findings, we wish to acknowledge several limitations in our work. The sample size was relatively small, which may limit the generalizability of our findings and result in us being unable to control for potential confounding factors that may have influenced our results. Nevertheless, we should stress that biopsy indication for suspected IPF patients is quite limited because of the high mortality as a serious complication of lung surgery within 30 days after the procedure, with an estimated frequency of 3–4% reported in most studies. Moreover, lung biopsy is only performed in difficult cases, which cannot be addressed by radiological and clinical evaluation, furtherly reducing the enlistable pool of patients. Finally, the study was conducted at a single center, which may limit the generalizability of our findings to other settings.

## 5. Conclusions

In conclusion, our study provides further evidence to support the use of fractal morphometry as a tool for quantifying and determining lung tissue remodeling in IPF. Our findings demonstrate a significant correlation between histological and radiological *D_f_* in the UIP pattern, as well as a significant association between *D_f_* and FF density. Furthermore, we determined the scalability and self-similarity of *D_f_* measurements across different biopsy types, including surgical and possibly cryobiopsy specimens, and we found an almost perfect correlation with retention of all correlation and survival parameters. This reinforces the viability of cryobiopsy as a valid alternative diagnostic tool to surgical biopsy.

## Figures and Tables

**Figure 1 biomedicines-11-01483-f001:**
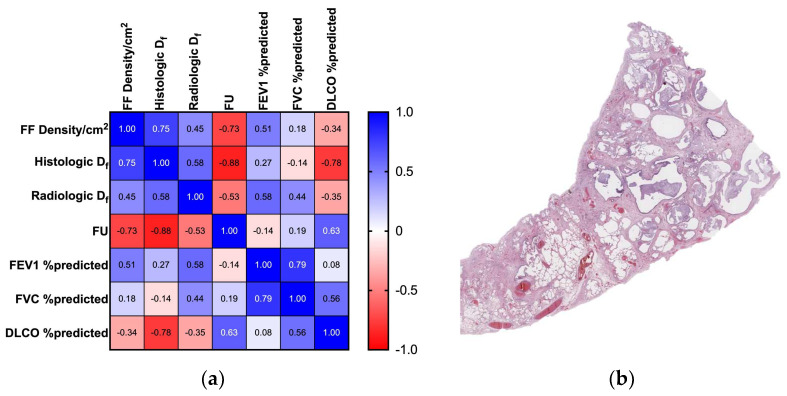
Correlation table and histologic depiction of UIP pattern: (**a**) This correlation table shows a summary of the main result in shadows of blue (positive correlation: 1 = perfect positive correlation) and red (negative correlation: −1 = perfect inverse correlation); (**b**) Histologic UIP pattern at low power, characterized by honeycombing, architectural distortion, and temporal inhomogeneity (1× original magnification, Hematoxylin and Eosin stain).

**Figure 2 biomedicines-11-01483-f002:**
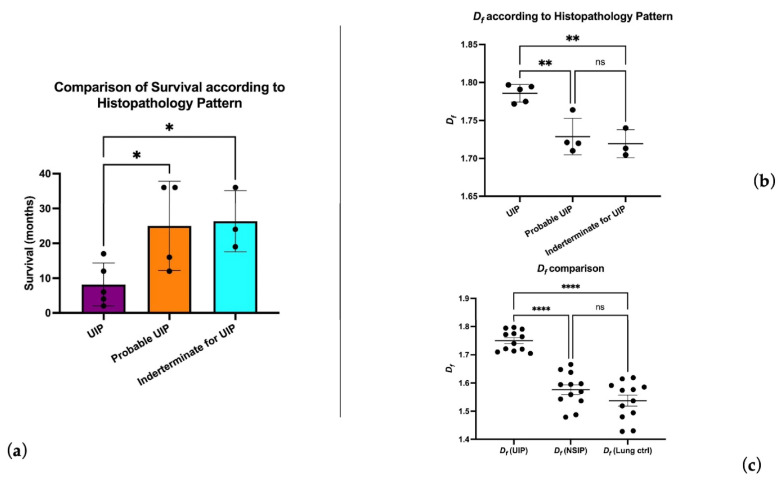
Fractal dimension analysis results: (**a**) *D_f_* in the UIP certain subgroup was associated with a shorter life expectancy; (**b**) *D_f_* was consistently higher in the UIP certain subgroup; (**c**) higher architectural distortion and complexity is confirmed by *D_f_* measurements compared to NSIP and control patients (ns = *p* > 0.05; * = *p* ≤ 0.05; ** = *p* ≤ 0.01; **** = *p* ≤ 0.0001).

**Figure 3 biomedicines-11-01483-f003:**
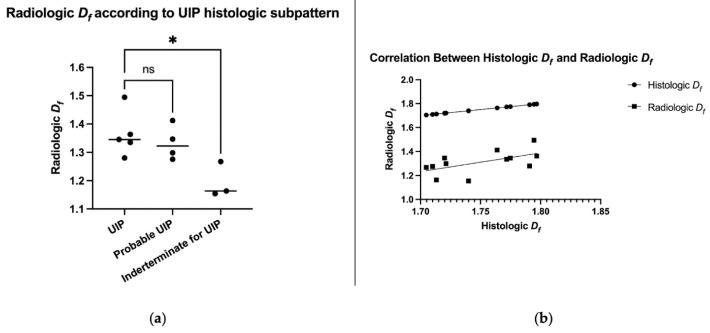
Role of radiologic fractal dimension: (**a**) a high degree of correlation was obtained between histologic specimens and radiologic images; (**b**) the *_f_* in CT scans of the UIP certain histologic pattern was significantly different from the UIP indeterminate patients (ns = *p* > 0.05; * = *p* ≤ 0.05).

**Figure 4 biomedicines-11-01483-f004:**
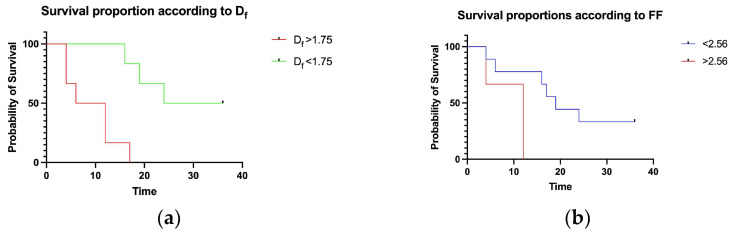
Survival proportion according to *D_f_* and FF density: (**a**) *D_f_* could actually identify a larger number of patients with unfavorable prognosis compared to (**b**) FF density by itself.

**Figure 5 biomedicines-11-01483-f005:**
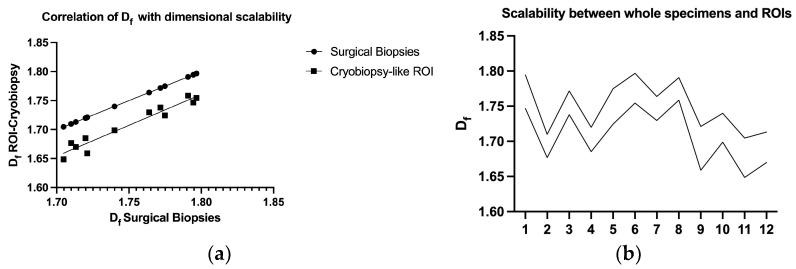
Scalability between the whole section image (~400 mm^2^) and the ROI, mimicking a cryobiopsy area (10–36 mm^2^): (**a**) an almost perfect correlation between the values was achieved (+0.97, r^2^ = 0.94); (**b**) consistent results among each specimen were high, despite a different UIP histologic pattern.

**Table 1 biomedicines-11-01483-t001:** Clinical data from UIP and control patients.

Clinical Data	UIP	Control
Age	73.4 ± 5.07	35.7 ± 12.3
Sex	M:11; F1	M:7; F5
Pack/years smoking	35.7 ± 7.8	0/occasional
FEV1 %predicted	73 ± 0.14	na
FVC %predicted	73 ± 0.17	na
DLCO %predicted	77 ± 0.04	na

**Table 2 biomedicines-11-01483-t002:** Histologic UIP pattern subgrouping and morphological parameters.

UIP Pattern	N= 5: Certain N= 4: Probable N = 3: Indeterminate
FF Density/cm^2^	2.538 ± 2.17
Histologic *D_f_*	1.75 ± 0.03
Radiologic *D_f_*	1.312 ± 0.09
Follow Up	18.33 ± 12.36

## Data Availability

The data presented in this study are available within the article.

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
