# Peer review of "Histologic Analysis of Idiopathic Pulmonary Fibrosis by Morphometric and Fractal Analysis"

_biomedicines, 2023, doi:10.3390/biomedicines11051483_

Round 1
Reviewer 1 Report
The manuscript by Mancini et al. used a rigorous methodology, including the box counting method on the FracLac ImageJ plugin, to evaluate Df and lacunarity measures. The study included 12 IPF patients, 12 NSIP patients, and 12 age-matched heathy control tissue from patients undergoing lung surgery for other causes. The study demonstrated the scalability and self-similarity of Df measurements across different biopsy types, which could have important implications for the diagnosis and management of IPF. However, the small sample size, retrospective design, and lack of information on potential confounding factors could limit the strength of the conclusions.
Comments:
1. A sample size of 12 patients may be considered small and may limit the statistical power of the study. Considerations could be made to increase the sample size, if possible.
2. Need details on data collection, such as how the data was collected and how the data was validated.
3. Imbalanced gender distribution: The sample includes 11 male patients and only 1 female patient, which may limit the generalizability of the findings to female patients.
4. A follow-up period of 36 months may not be sufficient to capture the long-term outcomes of UIP patients.
Author Response
We thank the Reviewer for Her/His suggestions.
Comments:
- A sample size of 12 patients may be considered small and may limit the statistical power of the study. Considerations could be made to increase the sample size, if possible.
We agree with the reviewer about the small sample size, nevertheless we should stress that biopsy indication for suspected IPF patients is quite limited because of the high mortality as serious complication of lung surgery within 30 days after the procedure, with an estimated frequency 3–4% reported in most studies. Moreover lung biopsy is only performed in difficult cases which cannot be addressed by radiological and clinical evaluation. We also implemented the text accordingly (line 286)
- Need details on data collection, such as how the data was collected and how the data was validated.
Morphometrical analyses was carried out by a dedicated pathologist who performed all the measurements blinded to group assignment, we made it more clear in text as well (line). Data validation is currently ongoing on smaller biopsies, and we wish it could be the subject for a future study.
- Imbalanced gender distribution: The sample includes 11 male patients and only 1 female patient, which may limit the generalizability of the findings to female patients.
We agree with the Reviewer there is in imbalance in gender distribution although recent studies have demonstrated a male predominance in IPF ranging in 70-80% (cfr. Sesé L, Nunes H, Cottin V, et al. Gender Differences in Idiopathic Pulmonary Fibrosis: Are Men and Women Equal?. Front Med (Lausanne). 2021;8:713698. Published 2021 Aug 5. doi:10.3389/fmed.2021.713698; Jo HE, Glaspole I, Grainge C, et al. Baseline characteristics of idiopathic pulmonary fibrosis: analysis from the Australian Idiopathic Pulmonary Fibrosis Registry [published correction appears in Eur Respir J. 2017 Mar 29;49(3):]. Eur Respir J. 2017;49(2):1601592. Published 2017 Feb 23. doi:10.1183/13993003.01592-2016) )
- A follow-up period of 36 months may not be sufficient to capture the long-term outcomes of UIP patients.
We thank the Reviewer for this observation. We should highlight that IPF prognosis is grim expecially if backed by certain histologic, radiologic, and clinical diagnosis. At the end of our 3 years follow-up only 3 out 12 patients were still alive (as stated at the beginning of the result section, line 152).
Reviewer 2 Report
Mancini and colleagues present a research article regarding the course of fibrotic changes in Idiopathic pulmonary fibrosis and interstitial pneumonia using histologic measurements. Here, they have established a protocol of how to detect significant changes at earlier stages which may also influence the diagnostic and therapeutic measures. The manuscript is well structured, the content is clear and the research methods are appropriate for this article. I have nothing significant concerns. There are some minor grammatical mistakes. The quality of the figures should be increased. Good job!
There are some minor grammatical mistakes.
Author Response
We thank the Reviewer for Her/His appreciation and comments.
High resolution images have been uploaded and grammatical mistakes have been double checked.
Reviewer 3 Report
this is a very interesting exploratory study.
two minor issues
1. please explain in more medical terms what this fractal morphometry means and which was initially applied. the way it reads in the draft is more for biophysicians. also please briefly comment on the feasibility of such a method in the routine work up
2. have you tried to correlated fibrosis pattern with neoplasia histology? please briefly comment in the draft if you find this issue meaningful
Author Response
We thank the Reviewer for Her/his suggestion.
1. please explain in more medical terms what this fractal morphometry means and which was initially applied. the way it reads in the draft is more for biophysicians. also please briefly comment on the feasibility of such a method in the routine work up
We agree with the Reviewer for the need of a better explanation of fractal morphometry and modified the text accordingly (line 70). We also added a brief paragraph on the feasibility of the method in the routine work up. (line 277)
2. have you tried to correlated fibrosis pattern with neoplasia histology? please briefly comment in the draft if you find this issue meaningful
We thank the reviewer for Her/His observation, we did not found any correlation between UIP pattern and lung cancer histology and amended the text subsequently (line 166).
Reviewer 4 Report
The presentation of the results should be improved (Tables, Figures, figure legends). Methods - the statistics need much more detailed explanation.
Line 44-46 - please reword incorporating - better detection, better analysis of pathogenesis and together led to development of new drugs
We do not know if the prevalence has truly increased
Next sentence, also needs quick editing
Table 1 - has empty columns
Figure 1: should indicate what the numbers mean
Table 2: needs to indicate what the numbers mean. Because none of the data reaches the value - certain or indeterminate, the data are hard to interpret
Figure 2a: follow up - should this be replaced by survival (months), or mortality (months)
English needs some careful editing
Author Response
We thank the reviewer for Her/His suggestions.
Line 44-46 - please reword incorporating - better detection, better analysis of pathogenesis and together led to development of new drugs
We acknowledge the reviewer’s comment and amended the text consequently.
We do not know if the prevalence has truly increased
Next sentence, also needs quick editing
We thank the reviewer for the suggestion and amended the text accordingly.
Table 1 - has empty columns
We acknowledge the reviewer’s comment and apologize for the mistake, the table has now been corrected.
Figure 1: should indicate what the numbers mean
We thank the reviewer for the suggestion and improved the figure legend.
Table 2: needs to indicate what the numbers mean. Because none of the data reaches the value - certain or indeterminate, the data are hard to interpret
We acknowledge the reviewer’s comment and modified Table 2 accordingly.
Figure 2a: follow up - should this be replaced by survival (months), or mortality (months)
We agree with the reviewer’s suggestion and modified Figure 2a.
Round 2
Reviewer 4 Report
thank you, the manuscript is much improved
improved
Author Response
We thank the Reviewer for Her/His acknowledgement.